# Compositional Generalization
# by Learning Analytical Expressions

**Qian Liu**[†*], **Shengnan An**[◇*], **Jian-Guang Lou**[§], **Bei Chen**[§], **Zeqi Lin**[§],
**Yan Gao**[§], **Bin Zhou**[†], **Nanning Zheng**[◇], **Dongmei Zhang**[§]
[†]Beihang University, Beijing, China;[◇]Xi'an Jiaotong University, Xi'an, China;
[§]Microsoft Research, Beijing, China
[†]{qian.liu, zhoubin}@buaa.edu.cn; [◇]{an1006634493@stu, nnzheng@mail}.xjtu.edu.cn;
[§]{jlou, beichen, Zeqi.Lin, Yan.Gao, dongmeiz}@microsoft.com

## Abstract

Compositional generalization is a basic and essential intellective capability of human beings, which allows us to recombine known parts readily. However, existing neural network based models have been proven to be extremely deficient in such a capability. Inspired by work in cognition which argues compositionality can be captured by variable slots with symbolic functions, we present a refreshing view that connects a memory-augmented neural model with analytical expressions, to achieve compositional generalization. Our model consists of two cooperative neural modules, Composer and Solver, fitting well with the cognitive argument while being able to be trained in an end-to-end manner via a hierarchical reinforcement learning algorithm. Experiments on the well-known benchmark SCAN demonstrate that our model seizes a great ability of compositional generalization, solving all challenges addressed by previous works with 100% accuracies.

## 1 Introduction

When using language, humans have a remarkable ability to recombine known parts to understand novel sentences they have never encountered before [8, 12]. For example, once humans have learned the meanings of "walk", "jump" and "walk twice", it is effortless for them to understand the meaning of "jump twice". This kind of ability relies on the compositionality that characterizes languages. The principle of compositionality refers to the idea that the meaning of a complex expression (e.g. a sentence) is determined by the meanings of its constituents (e.g. the verb "jump" and the adverb "twice") together with the way these constituents are combined (e.g. an adverb modifies a verb) [34]. Understanding language compositionality is a basic and essential capacity for human beings, which is argued to be one of the key skills towards human-like machine intelligence [25].

Recently, Lake and Baroni [19] made a step towards exploring and benchmarking compositional generalization of neural networks. They argued that leveraging compositional generalization was an essential ability for neural networks to understand out-of-domain sentences. The test suite, their proposed **S**implified version of the **C**ommAI **N**avigation (SCAN) dataset, contains compositional navigation commands, such as "walk twice", and corresponding action sequences, like WALK WALK. Such a task lies in the category of machine translation, and thus is expected to be well solved by current state-of-the-art translation models (e.g. sequence to sequence with attention [32, 3]). However, experiments on SCAN demonstrated that modern translation models dramatically fail to obtain a satisfactory performance on compositional generalization. For example, although the meanings of "walk", "walk twice" and "jump" have been seen, current models fail to generalize

---

[*] Work done during an internship at Microsoft Research. The first two authors contributed equally.

to understand "jump twice". Subsequent works verified that it was not an isolated case, since convolutional encoder-decoder model [10] and Transformer [17] met the same problem. There have been several attempts towards SCAN, but so far no neural based model can successfully solve all the compositional challenges on SCAN without extra resources [21, 18, 13].

In this paper, we propose a memory-augmented neural model to achieve compositional generalization by **L**earning **An**alytical **E**xpressions (LANE). Motivated by work in cognition which argues compositionality can be captured by variable slots with symbolic functions [4], our memory-augmented architecture is devised to contain two cooperative neural modules accordingly: Composer and Solver. Composer aims to find structured analytical expressions from unstructured sentences, while Solver focuses on understanding these expressions with accessing Memory (Sec. 3). These two modules are trained to learn analytical expressions together in an end-to-end manner via a hierarchical reinforcement learning algorithm (Sec. 4). Experiments on a well-known benchmark SCAN demonstrate that our model seizes a great ability of compositional generalization, reaching $100\%$ accuracies in all tasks (Sec. 5). As far as we know, our model is the first neural model to pass all compositional challenges addressed by previous works on SCAN without extra resources. We open-source our code at `https://github.com/microsoft/ContextualSP`.

## 2 Compositional Generalization Assessment

Since the study on compositional generalization of deep neural models is still in its infancy, the overwhelming majority of previous works employ artificial datasets to conduct assessment. As one of the most important benchmarks, the SCAN dataset is proposed to evaluate the compositional generalization ability of translation models [19]. As mentioned above, SCAN describes a simple navigation task that aims to translate compositional navigation sentences into executed action sequences. However, due to the open nature of compositional generalization, there is disagreement about which aspect should be addressed [34, 20, 15, 17]. To conduct a comprehensive assessment, we consider both *systematicity* and *productivity*, two important arguments for compositional generalization.

Systematicity evaluates if models can recombine known parts. To assess it, Lake and Baroni [19] proposed three tasks: (i) **Add Jump**. The pairs of train and test are split in terms of the primitive JUMP. All commands that contain, but are not exactly, the word "jump" form the test set. The rest forms the train set. (ii) **Around Right**. Any compositional command whose constitutes include "around right" is excluded from the train test. This task is proposed to evaluate whether the model can generalize the experience about "left" to "right", especially on "around right". (iii) **Length**. All commands with long outputs (i.e. output length is longer than 24), such as "around ∗ twice ∗ around" and "around ∗ thrice", are never seen in training, where "∗" indicates a wildcard. More recently, Keysers et al. [17] proposed another assessment, the distribution-based systematicity. It aims to measure compositional generalization by using a setup where there is a large compound distribution divergence between train and test sets (Maximum Compound Divergence, **MCD**) [17].

Productivity is thought to be another key argument. It not only requires models to recombine known parts, but also evaluates if they can productively generalize to inputs beyond the length they have seen in training. It relates itself to the unboundedness of languages, which means languages license a theoretically infinite set of possible sentences [4]. To evaluate it, we re-create the SCAN dataset (SCAN-ext). Compared with SCAN using up to one "and" in a sentence, SCAN-ext roughly controls the distribution of input lengths by the number of "and" (e.g. "jump and walk twice and turn left"). Input sentences in the train set consist of at most 2 "and", while the test set allows at most 9. Except for "and", the generation of other parts follows the procedure in SCAN.

## 3 Methodology

In this section, we first show the intrinsic connection between language compositionality and analytical expressions. We then describe how these expressions are learned through our model.

### 3.1 Problem Statement

Cognitive scientists argue that the compositionality of language indeed constitutes an algebraic system, of the sort that can be captured by variable slots with symbolic functions [4, 12]. As an

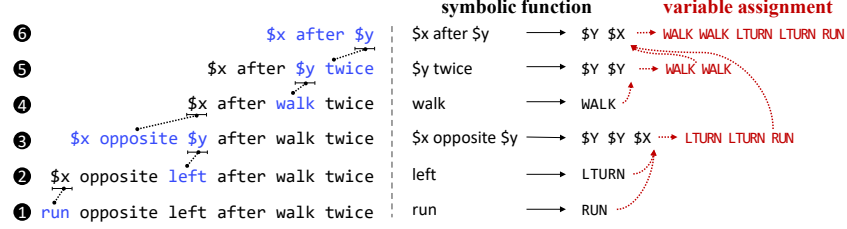

Figure 1: The schematic illustration of learning analytical expressions. The understanding of "run opposite left after walk twice" can be regarded as a hierarchical application of symbolic functions.

illustrative example, any adjective attached with the prefix "super-" can be regarded as applying a symbolic function (i.e. "super-adj") on a variable slot (e.g. "good"), and will be mapped to a new adjective (e.g. "super-good") [4]. Such a formulation frees the symbolic function from specific adjectives and makes it able to generalize on new adjectives (e.g. "super-bad").

Taking a more complicated case from SCAN, as shown in Fig. 1, "$x" and "$y" are variables defined in the source domain, and "$X" and "$Y" are variables defined in the destination domain. We call a sequence of source domain variables or words (e.g. run) a **source analytical expression** (SrcExp), while we call a sequence of destination domain variables or action words (e.g. RUN) a **destination analytical expression** (DstExp). If there is no variable in an SrcExp (or DstExp), it is also a **constant** SrcExp (or DstExp). From bottom to top, each phrase marked blue represents an SrcExp which will be superseded by a source domain variable (e.g. $x) when moving to the next hierarchy of understanding. These SrcExps can be recognized and translated into their corresponding DstExps by a set of symbolic functions. We call such SrcExps as **recognizable** SrcExps, and their corresponding DstExps as recognizable DstExps. By iterative recognizing and translating recognizable SrcExps, we can construct a tree hierarchy with a set of recognizable DstExps. By assigning values to the destination variables in recognizable DstExps recursively (dotted red arrows in Fig. 1), we can finally obtain a constant DstExp as the final resulted sequence.

It is well known that, variables are pieces of memory in computers, and a memory mechanism can be used to support variable-related operations. Thus we propose a memory-augmented neural model to achieve compositional generalization by automatically learning the above analytical expressions.

## 3.2 Model Design

Our model takes several steps to understand a sentence. Fig. 2 presents the overall procedure of our model at step $t$ and $t+1$ in detail (corresponding to step 5 and 6 in Fig. 1). At the beginning of step $t$,

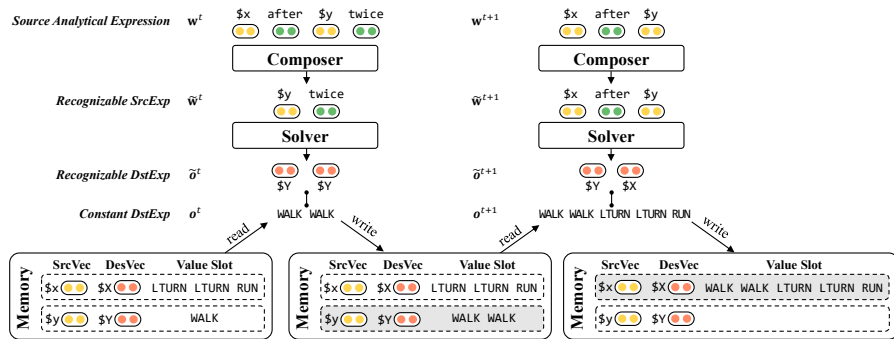

Figure 2: The illustration of our model. Colored neurons are learnable vectors. Composer accepts an SrcExp as input, and aims to find a recognizable SrcExp inside it. Solver first translates a recognizable SrcExp into a recognizable DstExp, and then assigns values to destination variables in the recognizable DstExp, obtaining a constant DstExp. To support variable-related operations in a differentiable manner [31], Memory is designed to include a number of items, each of which contains a source vector (SrcVec) to represent source variables (e.g. $x, $y), a destination vector (DesVec) to represent destination variables (e.g. $X, $Y), and a value slot to temporarily store a constant DstExp.

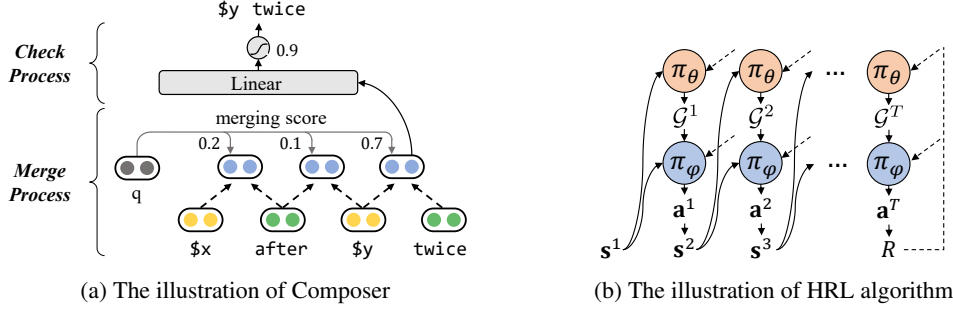

| (a) The illustration of Composer | (b) The illustration of HRL algorithm |

Figure 3: (a) Composer finds a recognizable SrcExp via the cooperation of the merge process and the check process. (b) Our HRL algorithm contains a high-level policy $\pi_\theta$ and a low-level policy $\pi_\varphi$.

an SrcExp "\$x after \$y twice" is fed into Composer. Then Composer finds a recognizable SrcExp "\$y twice" and sends it to Solver. Receiving "\$y twice", Solver first translates it into "\$Y \$Y". Using "\$Y \$Y" as the skeleton, Solver obtains WALK WALK by replacing "\$Y" with its corresponding constant DstExp WALK stored in Memory. Meanwhile, since WALK has been used, the value slot which stores WALK is set to empty. Next, Solver applies for one item with an empty value slot in Memory, i.e. the item containing \$y and \$Y, and then writes WALK WALK into its value slot (gray background in Fig.2). Finally, the recognizable SrcExp "\$y twice" in $\mathbf{w}^t$ is superseded by "\$y", producing "\$x after \$y" as $\mathbf{w}^{t+1}$ for the next step. Such a procedure is repeated until the SrcExp fed into Composer is a recognizable SrcExp. Assuming the step at this point is $T$, the constant DstExp $\mathbf{o}^T$ is actually the final output action sequence.

**Composer** Given an SrcExp $\mathbf{w}^t$, Composer aims to find a recognizable SrcExp $\tilde{\mathbf{w}}^t$. There are several ways to implement it, and we choose to gradually merge elements of $\mathbf{w}^t$ until a recognizable SrcExp appears. As shown in Fig. 3a, given "\$x after \$y twice", at first Composer merges "\$y" and "twice". Then it checks if "\$y twice" is a recognizable SrcExp. In this case the answer is YES, and thus Composer triggers Solver to translate "\$y twice". Otherwise, the overall procedure would be iterative, which means that Composer would continue to merge until a recognizable SrcExp appears. Viewing the procedure as building a binary tree from bottom to top, Composer iteratively merges two neighboring elements of $\mathbf{w}^t$ into a parent node at each layer (i.e. the *merge* process), and checks if the parent node represents a recognizable SrcExp (i.e. the *check* process).

The merge process is implemented by first enumerating all possible parent nodes of the current layer, and then selecting the one which has the highest merging score. Assuming $i$-th and $(i+1)$-th node at layer $l$ are represented by $\mathbf{r}_i^l$ and $\mathbf{r}_{i+1}^l$ respectively, their parent representation $\mathbf{r}_i^{l+1}$ can be obtained via a standard Tree-LSTM encoding [35] using $\mathbf{r}_i^l$ and $\mathbf{r}_{i+1}^l$ as input. As shown in Fig. 3a, given all parent node representations (blue neurons), Composer selects the parent node (solid lines with arrows) whose merging score is the maximum. In fact, the merging score measures the merging priority of $\mathbf{r}_i^{l+1}$ using a learnable query vector $\mathbf{q}$ by $\langle \mathbf{q}, \mathbf{r}_i^{l+1} \rangle$, where $\langle \cdot, \cdot \rangle$ represents the inner product. Once the parent node for layer $l$ is determined, the check process begins.

The check process is to check if a parent node represents a recognizable SrcExp. Concretely, denoting $\mathbf{r}_i^{l+1}$ the parent node representation, an affine transformation is built based on it to obtain the probability $p_c = \sigma(\mathbf{W}_c \mathbf{r}_i^{l+1} + b_c)$ where $\mathbf{W}_c$ and $b_c$ are learned parameters and $\sigma$ is the sigmoid function. $p_c > 0.5$ means that the parent node represents a recognizable SrcExp, and thus Composer triggers Solver to translate it. Otherwise, the parent node and other unmerged nodes enter a new layer $l+1$, based on which Composer restarts the merge process.

**Solver** Given a recognizable SrcExp $\tilde{\mathbf{w}}^t$, Solver first translates it into a recognizable DstExp $\tilde{\mathbf{o}}^t$, and then obtains a constant DstExp $\mathbf{o}^t$ via variable assignment through interacting with Memory. To achieve this, Solver is designed to be an LSTM-based sequence to sequence network with an attention mechanism [3]. It generates the recognizable DstExp via decoding it step by step. At each step, Solver either generates an action word, or a destination variable. Using the recognizable DstExp as the skeleton, Solver obtains a constant DstExp by replacing each destination variable with its corresponding constant DstExp stored in Memory.

# 4 Model Training

Training our proposed model is non-trivial for two reasons: (i) since the identification of $\tilde{\mathbf{w}}^t$ is discrete, it is hard to optimize Composer and Solver via back propagation; (ii) since there is no supervision about $\tilde{\mathbf{w}}^t$ and $\tilde{\mathbf{o}}^t$, Composer and Solver cannot be trained separately. Recalling the procedure of these two modules in Fig. 2, it is natural to model the problem via Hierarchical Reinforcement Learning (HRL) [5]: a high-level agent to find recognizable SrcExps (Composer), and a low-level agent to obtain constant DstExps conditioned on these recognizable SrcExps (Solver).

## 4.1 Hierarchical Reinforcement Learning

We begin by introducing some preliminary formulations for our HRL algorithm. Denoting $\mathbf{s}^t$ as the **state** at step $t$, it contains both $\mathbf{w}^t$ and Memory. The **action** of Composer, denoted by $\mathcal{G}^t$, is the recognizable SrcExp to be found at step $t$. Given $\mathbf{s}^t$ as observation, the parameter of Composer $\theta$ defines a **high-level policy** $\pi_\theta(\mathcal{G}^t \mid \mathbf{s}^t)$. Once a high-level action $\mathcal{G}^t$ is produced, the low-level agent Solver is triggered to react following a **low-level policy** conditioned on $\mathcal{G}^t$. In this sense, the high-level action can be viewed as a sub-goal for the low-level agent. Denoting $\mathbf{a}^t$ the action of Solver, the low-level policy $\pi_\varphi(\mathbf{a}^t \mid \mathcal{G}^t, \mathbf{s}^t)$ is parameterized by the parameter of Solver $\varphi$. $\mathbf{a}^t$ is the constant DstExp output by Solver at step $t$. More implementation details about $\pi_\theta$ and $\pi_\varphi$ can be found in the supplementary material.

**Policy Gradient**   As illustrated in Fig. 3b, in our HRL algorithm, Composer and Solver take actions in turn. When it is Composer's turn to act, it picks a sub-goal $\mathcal{G}^t$ according to $\pi_\theta$. Once $\mathcal{G}^t$ is set, Solver is triggered to pick a low-level action $\mathbf{a}^t$ according to $\pi_\varphi$. These two modules alternately act until they reach the endpoint (i.e. step $T$) and predict the output action sequence, forming a trajectory $\tau = (\mathbf{s}^1 \mathcal{G}^1 \mathbf{a}^1 \cdots \mathbf{s}^T \mathcal{G}^T \mathbf{a}^T)$. Once $\tau$ is determined, the reward is collected to optimize $\theta$ and $\varphi$ using policy gradient [33]. Denoting $R(\tau)$ as the reward of a trajectory $\tau$ (elaborated in Sec. 4.2), the training objective of our model is to maximize the expectation of rewards as:

$$\max_{\theta,\varphi} \mathcal{J}(\theta,\varphi) = \max_{\theta,\varphi} \mathbb{E}_{\tau \sim \pi_{\theta,\varphi}} R(\tau). \tag{1}$$

Applying the likelihood ratio trick, $\theta$ and $\varphi$ can be optimized by ascending the following gradient:

$$\nabla_{\theta,\varphi} \mathcal{J}(\theta,\varphi) = \mathbb{E}_{\tau \sim \pi_{\theta,\varphi}} R(\tau) \nabla_{\theta,\varphi} \log \pi_{\theta,\varphi}(\tau). \tag{2}$$

Expanding the above equation via the chain rule[2], we can obtain:

$$\nabla_{\theta,\varphi} \mathcal{J}(\theta,\varphi) = \mathbb{E}_{\tau \sim \pi_\theta} \sum_t R(\tau) \left[ \nabla_{\theta,\varphi} \log \pi_\theta \left( \mathcal{G}^t | \mathbf{s}^t \right) + \nabla_{\theta,\varphi} \log \pi_\varphi \left( \mathbf{a}^t | \mathcal{G}^t, \mathbf{s}^t \right) \right]. \tag{3}$$

Considering the search space of $\tau$ is huge, the REINFORCE algorithm [39] is leveraged to approximate Eq. 3 by sampling $\tau$ from $\pi_{\theta,\varphi}$ for $N$ times. Furthermore, the technique of subtracting a baseline [38] is employed to reduce variance, where the baseline is the mean reward over sampled $\tau$.

**Differential Update**   Unlike standard Reinforcement Learning (RL) algorithms, we introduce a *differential update* strategy to optimize Composer and Solver via different learning rates. It is motivated by an intuition that actions of a high-level agent cannot be changed quickly. According to Eq. 3, simplifying $\mathbb{E}_{\tau \sim \pi_{\theta,\varphi}}$ as $\mathbb{E}$, the parameters of Composer and Solver are optimized as:

$$\theta \leftarrow \theta + \alpha \cdot \mathbb{E} \, R(\tau) \sum_t \nabla_\theta \log \pi_\theta \left( \mathcal{G}^t | \mathbf{s}^t \right), \quad \varphi \leftarrow \varphi + \beta \cdot \mathbb{E} \, R(\tau) \sum_t \nabla_\varphi \log \pi_\varphi \left( \mathbf{a}^t | \mathcal{G}^t, \mathbf{s}^t \right), \tag{4}$$

where Solver's learning rate $\beta$ is greater than Composer's learning rate $\alpha$.

## 4.2 Reward Design

The design of the reward function is critical to an RL based algorithm. Bearing this in mind, we design our reward from two aspects: similarity and simplicity. It is worth noting that both rewards work globally, i.e., all actions share the same reward, as indicated by dotted lines in Fig. 3b.

**Similarity-based Reward**    It is based on the similarity between the model's output and the ground-truth. Since the output of our model is an action sequence, a kind of sequence similarity, the Intersection over Union (**IoU**) similarity, is employed as the similarity-based reward function. Given the sampled output $\mathbf{a}^T$ and the ground-truth $\mathbf{o}$, the similarity-based reward is computed by:

$$R_{\mathrm{s}}(\tau) = \left| \mathbf{a}^T \cap \mathbf{o} \right| / \left( \left| \mathbf{a}^T \right| + |\mathbf{o}| - \left| \mathbf{a}^T \cap \mathbf{o} \right| \right), \tag{5}$$

where $\mathbf{a}^T \cap \mathbf{o}$ means the longest common substring between $\mathbf{a}^T$ and $\mathbf{o}$, and $| \cdot |$ represents the length of a sequence. Compared with exact matching, such a reward alleviates the reward sparsity issue.

**Simplicity-based Reward**    Inspired by Occam's Razor principle that "the simplest solution is most likely the right one", we try to encourage our model to have the fewest kinds of learned recognizable DstExps overall. In other words, we encourage the model to fully utilize variables and be more generalizable. Taking an illustration of "jump twice", $[\,\text{jump twice} \rightarrow \text{JUMP JUMP}\,]$ and $[\,\text{jump} \rightarrow \text{JUMP}, \$x \text{ twice} \rightarrow \$X \$X\,]$ both result in correct outputs. Intuitively, the latter is more generalizable as it enables Solver to reuse learned recognizable DstExps, more in line with the Occam's Razor principle. Concretely, when understanding a novel input like "walk twice", $\$x$ twice $\rightarrow \$X \$X$ can be reused. Denoting $T^*$ as the number of steps where the recognizable DstExp only contains destination variables, we design a reward $R_{\mathrm{a}}(\tau) = T^* / T$ as a measure of the simplicity. The final reward function $R(\tau)$ is a linear summation as $R(\tau) = R_{\mathrm{s}}(\tau) + \gamma \cdot R_{\mathrm{a}}(\tau)$, where $\gamma$ is a hyperparameter.

## 4.3  Curriculum Learning

One typical strategy for improving model generalization capacity is to use curriculum learning, which arranges examples from easy to hard in training [24, 1]. Inspired by it, we divide the training into different lessons according to the length of the input sequence. Our model starts training on the simplest lesson, with lesson complexity gradually increasing. Besides, as done in literature [7], we accumulate training data from previous lessons to avoid catastrophic forgetting.

# 5  Experiments

In this section, we conduct a series of experiments to evaluate our model on various compositional tasks mentioned in Sec. 2. We then verify the importance of each component via a thorough ablation study. Finally we present two real cases to illustrate our model concretely.

## 5.1  Experimental Setup

**Task**    In this section, we introduce *Tasks* used in our experiments. Systematicity is evaluated on *Add Jump*, *Around Right* and *Length* of SCAN [19], while distribution-based systematicity is assessed on *MCD* splits of SCAN [17]. MCD uses a nondeterministic algorithm to split examples into the train set and the test set. By using different random seeds, it introduces three tasks *MCD1*, *MCD2*, and *MCD3*. Productivity is evaluated on the SCAN-ext dataset. In addition, we also conduct experiments on the *Simple* task of SCAN which requires no compositional generalization capacity, and the *Limit* task of MiniSCAN [20] which evaluates if models can learn compositional generalization when given limited (i.e. 14) training data. We follow previous works to split datasets for all tasks.

**Baselines**    We consider a range of state-of-the-art models on SCAN compositional tasks as our baselines. In terms of the usage of extra resources, we divide them into two groups: (i) **No Extra Resources** includes vanilla sequence to sequence with attention (Seq2Seq) [19, 23], convolutional sequence to sequence (CNN) [10], Transformer [36], Universal Transformer [9], Syntactic Attention [29] and Compositional Generalization for Primitive Substitutions (CGPS) [21]. (ii) **Using Extra Resources** consists of Good Enough Compositional Data Augmentation (GECA) [2], meta sequence to sequence (Meta Seq2seq) [18], equivariant sequence to sequence (Equivariant Seq2seq) [13] and Program Synthesis [27]. Here we define "extra resources" as "data or data specific rules other than original training data". GECA and Meta Seq2Seq lie in "extra resources" since they both utilize extra data. Specifically, GECA recombines real examples to construct extra data, while Meta Seq2Seq employs random assignment of the primitive instructions (e.g. "jump") to their meaning (e.g. JUMP) to synthesize extra data. Regarding data-specific rules, Equivariant Seq2Seq requires predefined local

Table 1: Test accuracies of systematicity assessment on the SCAN dataset. All results of LANE are obtained by averaging over 5 runs, the same for Tab. 2 and Tab. 3.

| Extra Resources | Model | Simple | Add Jump | Around Right | Length |
|---|---|---|---|---|---|
| None | Seq2Seq [19, 23] | 99.7 | 1.2 | 2.5± 2.7 | 13.8 |
| | CNN [10] | 100.0 | 69.2± 9.2 | 56.7±10.2 | 0.0 |
| | Syntactic Attention [29] | 100.0 | 91.0±27.4 | 28.9±34.8 | 15.2±0.7 |
| | CGPS [21] | 99.9 | 98.8± 1.4 | 83.2±13.2 | 20.3±1.1 |
| | LANE (Ours) | 100.0 | 100.0 | 100.0 | 100.0 |
| Data Augmentation | GECA [2] | - | 87.0 | 82.0 | - |
| Permutation-based Augmentation | Meta Seq2Seq [18] | - | 99.9 | 99.9 | 16.6 |
| Manually Designed Local Groups | Equivariant Seq2Seq [13] | 100.0 | 99.1± 0.0 | 92.0± 0.2 | 15.9±3.2 |
| Manually Designed Meta Grammar | Program Synthesis [27] | 100.0 | 100.0 | 100.0 | 100.0 |

Table 2: Test accuracies of the distribution-based systematicity assessment on the SCAN dataset (**left**) and the *Limit* task on the MiniSCAN dataset (**right**).

| Model | MCD1 | MCD2 | MCD3 |
|---|---|---|---|
| Seq2Seq [17] | 6.5±3.0 | 4.2±1.4 | 1.4±0.2 |
| Transformer [17] | 0.4±0.2 | 1.6±0.3 | 0.8±0.4 |
| Universal Transformer [17] | 0.5±0.1 | 1.5±0.2 | 1.1±0.4 |
| CGPS | 1.2±1.0 | 1.7±2.0 | 0.6±0.3 |
| LANE (Ours) | 100.0 | 100.0 | 100.0 |

| Model | Limit |
|---|---|
| Human [20] | 84.3 |
| Seq2Seq | 2.5 |
| CGPS | 76.0 |
| Meta Seq2Seq | 100.0 |
| LANE (Ours) | 100.0 |

groups to make the model aware of equivariance between verbs or directions (e.g. "jump" and "run" are verbs). Similarly, Program Synthesis needs a predefined meta grammar, which heavily relies on the grammar of a dataset, and hence we think it also falls into the group of using "extra resources". Details of these baselines can be found in Sec. 6.

## 5.2 Implementation Details

Our model is implemented in PyTorch [28]. All experiments use the same hyperparameters. Dimensions of word embeddings, hidden states, key vectors and value vectors are set as $128$. Hyperparameters $\gamma$ and $N$ are set as $0.5$ and $10$ respectively. All parameters are randomly initialized and updated via the AdaDelta [40] optimizer, with a learning rate of $0.1$ for Composer and $1.0$ for Solver. Meanwhile, as done in previous works [14], we introduce a regularization term to prevent our model from overfitting in the early stage of training. Its weight is set to $0.1$ at the beginning, and exponentially anneals with a rate $0.5$ as the lesson increases. Our model is trained on a single Tesla-P100 (16GB) and the training time for a single run is about $20 \sim 25$ hours.

## 5.3 Experimental Results

**Experiment 1: Systematicity on SCAN**  As shown in Tab. 1, LANE achieves stunning 100% test accuracies on all tasks. Compared with state-of-the-art baselines without extra resources, LANE achieves a significantly higher performance. Even compared to baselines with extra resources, LANE is highly competitive, suggesting that to some extent LANE is capable of learning human prior knowledge. Although program synthesis [27] also achieves perfect accuracies, it heavily depends on a predefined meta-grammar where decent task-related knowledge is encoded. As far as we know, LANE is the first neural model to pass all tasks without extra resources.

**Experiment 2: Distribution-based Systematicity on SCAN**  LANE also achieves 100% accuracies on the more challenging distribution-based systematicity tasks (see Tab. 2). By comparing Tab. 1 and Tab. 2, one can find LANE maintains a stable and perfect performance regardless of the task, while a strong baseline CGPS shows a sharp drop. Furthermore, to the best of our knowledge, LANE is also the first one to pass the assessment of distribution-based systematicity on SCAN.

**Experiment 3: Productivity**  As shown in Fig. 4a, there is a sharp divergence between input lengths of train and test set on SCAN-ext, suggesting it is a feasible benchmark for productivity. From the results (right), one can find that test accuracies of baselines are mainly ruled by the frequency of input lengths in the train set. In contrast, LANE maintains a perfect trend as the input length

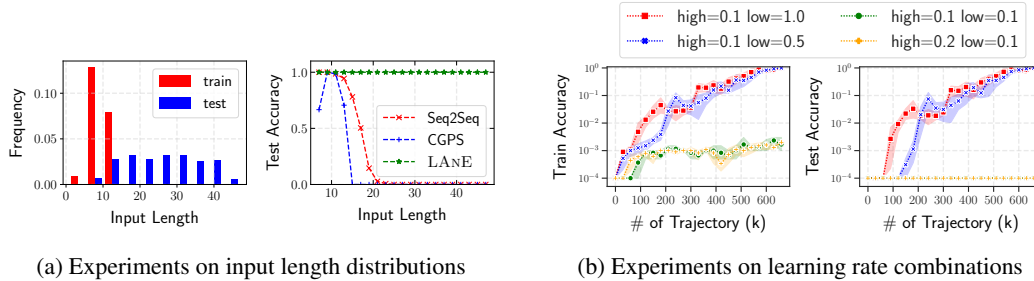

|  | (a) Experiments on input length distributions | (b) Experiments on learning rate combinations |

Figure 4: (a) Input length distributions on train set and test set of SCAN-ext (**left**) and test accuracies of various method on different input lengths (**right**). (b) Accuracies on train set (**left**) and test set (**right**) under different learning rate combinations.

Table 3: Test accuracies of different variants in all tasks on the SCAN dataset.

| Variant | Simple | Add Jump | Length | Around Right | MCD1 | MCD2 | MCD3 |
|---|---|---|---|---|---|---|---|
| w/o Composer | $98.5 \pm 0.6$ | 0.0 | $11.1 \pm 13.1$ | 0.0 | $5.3 \pm 2.4$ | $0.7 \pm 0.3$ | $2.6 \pm 0.9$ |
| w/o Curriculum Learning | 0.0 | 0.0 | 0.0 | 0.0 | 0.0 | 0.0 | 0.0 |
| w/o Simplicity-based Reward | 100.0 | 100.0 | 100.0 | 0.0 | 100.0 | 100.0 | $78.8 \pm 4.2$ |

increases, indicating it has productive generalization capabilities. Furthermore, the trend suggests the potential of LANE on tackling inputs with unbounded length.

**Experiment 4: Compositional Generalization on MiniSCAN**   Tab. 2 (**right**) shows the performance of various methods given limited training data, and LANE remains highly effective. Without extra resources such as permutation-based augmentation employed by Meta Seq2Seq, our model performs perfectly, i.e. 100% on the *Limit* task. Compared with the human performance $84.3\%$ [20], to a certain extent, our model is close to the human ability at learning compositional generalization from few examples. However, it does not imply that either our model or Meta Seq2Seq triumphs over humans in terms of compositional generalization, as the *Limit* task is relatively simple.

### 5.4   Closer Analysis

We conduct a thorough ablation study in Tab. 3 to verify the effectiveness of each component in our model. "w/o Composer" ablates the check process of Composer, making our model degenerate into a tree to sequence model, which employs a Tree-LSTM to build trees and encode input sequences dynamically. "w/o Curriculum Learning" means training our model on the full train set from the beginning. As the result shows, ablating each of above causes an enormous performance drop, indicating the necessity of Composer and the curriculum learning. Especially, without the curriculum learning, our model shows no sign of convergence even after training for several days, and thus all results are directly dropped to 0. We suppose that our model shows such non-convergence since its action space is exponentially large, which is due to the indefinite length of output sequences in Solver, and the huge number of possible trees in Composer. Such a huge space means that rewards are very sparse, especially for harder examples. So without curriculum learning, our randomly initialized model receives zero rewards on most examples. In comparison, by arranging examples from easy to hard, curriculum learning alleviates the sparse reward issue. On the one hand, easy examples are more likely to provide non-zero rewards to help our model converge; on the other hand, models trained on easy examples have a greater possibility to receive non-zero rewards on hard examples. "w/o Simplicity-based Reward", which only considers the similarity-based reward, fails on several tasks such as *Around Right*. We attribute its failure to its inability to learn sufficiently general recognizable DstExps from the data. As for the differential update, we compare the results of several learning rate combinations in Fig. 4b. As indicated, our designed differential update strategy is essential for successful convergence and high test accuracies. Last, we present learned tree structures of two real cases in Fig. 5. Observing that "twice" behaves differently under different contexts, it is non-trivial to produce such trees.

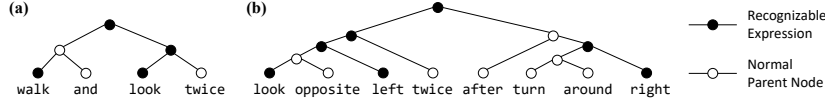

Figure 5: Learned tree structures in Composer of two real cases.

## 6   Related Work

The most related work is the line of exploring compositional generalization on neural networks, which has attracted a large attention on different topics in recent years. Under the topic of mathematical reasoning, Veldhoen et al. [37] explored the algebraic compositionality of neural networks via simple arithmetic expressions, and Saxton et al. [30] pushed the area forward by probing if the standard Seq2Seq model can resolve complex mathematical problems. Under the topic of logical inference, previous works devoted to testing the ability of neural networks on inferring logical relations between pairs of artificial language utterances [6, 26]. Our work differently focuses more on the compositionality in languages, benchmarked by the SCAN compositional tasks [19].

As for the SCAN compositional tasks, there have been several attempts. Inspired by work in neuro-science which suggests a disjoint processing on syntactic and semantic, Russin et al. [29] proposed the Syntactic Attention model. Analogously, Li et al. [21] employed different representations for primitives and functions respectively (CGPS). Unlike their separate representations, our proposed Composer and Solver can be seen as separate at the module level. There are also some works which impose prior knowledge of compositionality via extra resources. Andreas [2] presented a data augmentation technique to enhance standard approaches (GECA). Lake [18] argued to achieve compositional generalization by meta learning, and thus they employed a Meta Seq2Seq model with a memory mechanism. Regarding the memory mechanism, our work is similar to theirs. However, their training process, namely permutation training, requires handcrafted data augmentation. In a follow-up paper [27], they argued to generalize via the paradigm of program synthesis. Despite the nearly perfect performance, it also requires a predefined meta-grammar, where decent knowledge is encoded. Meanwhile, based on the group-equivariance theory, Gordon et al. [13] predefined local groups to enable models aware of equivariance between verbs or directions (Equivariant Seq2Seq). The biggest difference between our work and theirs is that we do not utilize any extra resource.

Our work is also related to those which apply RL on language. In this sense, using language as the abstraction for HRL [16] is the most related work. They proposed to use sentences as the sub-goal for the low-level policy in vision-based tasks, while we employ recognizable SrcExps as the sub-goal. In addition, the applications of RL on language involves topics such as natural language generation [11], conversational semantic parsing [22] and text classification [41].

## 7   Conclusion & Future Work

In this paper, we propose to achieve compositional generalization by learning analytical expressions. Motivated by work in cognition, we present a memory-augmented neural model which contains two cooperative neural modules Composer and Solver. These two modules are trained in an end-to-end manner via a hierarchical reinforcement learning algorithm. Experiments on a well-known benchmark demonstrate that our model solves all challenges addressed by previous works with 100% accuracies, surpassing existing baselines significantly. For future work, we plan to extend our model to a recently proposed compositional task CFQ [17] and more realistic applications.

## Acknowledgments

We thank all the anonymous reviewers for their valuable comments. This work was supported in part by National Natural Science Foundation of China (U1736217 and 61932003), and National Key R&D Program of China (2019YFF0302902).

## Broader Impact

This work explores the topic of compositional generalization capacities in neural networks, which is a fundamental problem in artificial intelligence but not involved in real applications at now. Therefore, there will be no foreseeable societal consequences nor ethical aspects.

## Footnotes

[2]More details can be found in the supplementary material.

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
