[Supplementary Material]

## A  Tree-LSTM Encoding

As mentioned in Sec. 3, a Tree-LSTM [35] model is employed to accomplish the merge process in Composer. Similar to LSTM, Tree-LSTM uses gate mechanisms to control the flow of information from child nodes to parent nodes. Meanwhile, it maintains a hidden state and a cell state analogously. Denoting $\mathbf{r}_i^l$ as the node representation of $i$-th node at layer $l$, it consists of the hidden state vector $\mathbf{h}_i^l$ and the cell state vector $\mathbf{c}_i^l$. For any parent node, its node representation $\mathbf{r}_i^l$ ($l > 1$) can be obtained by merging its left child node representation $\mathbf{r}_i^{l-1} = (\mathbf{h}_i^{l-1}, \mathbf{c}_i^{l-1})$ and right child node representation $\mathbf{r}_{i+1}^{l-1} = (\mathbf{h}_{i+1}^{l-1}, \mathbf{c}_{i+1}^{l-1})$ as:

$$
\begin{bmatrix} \mathbf{o} \\ \mathbf{f}_i^{l-1} \\ \mathbf{f}_{i+1}^{l-1} \\ \mathbf{e} \\ \mathbf{g} \end{bmatrix} = \begin{bmatrix} \sigma \\ \sigma \\ \sigma \\ \sigma \\ \tanh \end{bmatrix} \left( \mathbf{W}_{\text{tree}} \begin{bmatrix} \mathbf{h}_i^{l-1} \\ \mathbf{h}_{i+1}^{l-1} \end{bmatrix} + \mathbf{b}_{\text{tree}} \right),
$$
$$
\mathbf{c}_i^l = \mathbf{f}_i^{l-1} \odot \mathbf{c}_i^{l-1} + \mathbf{f}_{i+1}^{l-1} \odot \mathbf{c}_{i+1}^{l-1} + \mathbf{e} \odot \mathbf{g},
$$
$$
\mathbf{h}_i^l = \mathbf{o} \odot \tanh\left(\mathbf{c}_i^l\right), \tag{6}
$$

where $\mathbf{W}_{\text{tree}} \in \mathbb{R}^{5D_h \times 2D_h}$ is a learnable matrix, $\mathbf{b}_{\text{tree}} \in \mathbb{R}^{5D_h}$ is a learnable vector, $\sigma$ and $\tanh$ are activation functions, and $\odot$ represents the element-wise product. As for leaf nodes, their representations $\mathbf{r}_i^l$ ($l = 1$) can be obtained by applying leaf transformation on the embeddings of their corresponding elements $\mathbf{w}_i^t$ (e.g. \$x, after) as:

$$
\mathbf{r}_i^1 = \begin{bmatrix} \mathbf{h}_i^1 \\ \mathbf{c}_i^1 \end{bmatrix} = \mathbf{W}_{\text{leaf}} \operatorname{Emb}\left(\mathbf{w}_i^t\right) + \mathbf{b}_{\text{leaf}}, \tag{7}
$$

where $\mathbf{W}_{\text{leaf}} \in \mathbb{R}^{2D_h \times D_h}$ is a learnable matrix, $\mathbf{b}_{\text{leaf}} \in \mathbb{R}^{2D_h}$ is a learnable vector, $\mathbf{w}_i^t$ is the $i$-th element of $\mathbf{w}^t$, and $\operatorname{Emb}(\mathbf{w}_i^t) \in \mathbb{R}^{D_h}$ represents the word embedding if $\mathbf{w}_i^t$ is a word, otherwise the key vector of the source domain variable $\mathbf{w}_i^t$.

## B  Details about Policy

In the following, we will explain the high-level policy $\pi_\theta$ and the low-level policy $\pi_\varphi$ in detail. For the sake of clarity, we simplify $\mathbf{s}^t$, $\mathcal{G}^t$ and $\mathbf{a}^t$ as $\mathbf{s}$, $\mathcal{G}$ and $\mathbf{a}$, respectively.

**High-level policy**  Given $\mathbf{s}$, the high-level agent picks $\mathcal{G}$ according to the high-level policy $\pi_\theta(\mathcal{G} \mid \mathbf{s})$ parameterized by $\theta$. As mentioned in Sec. 3, $\mathcal{G}$ is obtained by applying in turn the merge and check process. Denoting the decisions made in the merge and check process at layer $l$ as $\mathcal{M}_l$ and $\mathcal{C}_l$, they are governed by parameters $\theta_\mathcal{M}$ and $\theta_\mathcal{C}$, respectively. A high-level action $\mathcal{G}$ is indeed a sequence of $\mathcal{M}$ and $\mathcal{C}$ as $(\mathcal{M}_1 \mathcal{C}_1 \cdots \mathcal{M}_L \mathcal{C}_L)$, where $L$ represents the highest layer. Therefore, $\pi_\theta(\mathcal{G} \mid \mathbf{s})$ is expanded as:

$$
\pi_\theta\left(\mathcal{G} = (\mathcal{M}_1 \mathcal{C}_1 \cdots \mathcal{M}_L \mathcal{C}_L) \mid \mathbf{s}\right) = \prod_{l=1}^{L} \pi_{\theta_\mathcal{M}}\left(\mathcal{M}_l \mid \mathbf{s}, \mathcal{M}_{<l}, \mathcal{C}_{<l}\right) \pi_{\theta_\mathcal{C}}\left(\mathcal{C}_l \mid \mathbf{s}, \mathcal{M}_{<l+1}, \mathcal{C}_{<l}\right), \tag{8}
$$

where $\pi_{\theta_\mathcal{M}}$ is implemented by a Tree-LSTM with a learnable query vector $\mathbf{q}$ (mentioned in Sec. 3.2). Assuming there are $K$ parent node candidates for layer $l$, $\mathcal{M}^l$ is a one-hot vector drawn from a $K$-dimensional categorical distribution $\pi_{\theta_\mathcal{M}}\left(\mathcal{M}_l \mid \mathbf{s}, \mathcal{M}_{<l}, \mathcal{C}_{<l}\right)$ with the weight $(p_1, \cdots, p_K)$. For the $k$-th parent node candidate, represented by $\mathbf{r}_k^{l+1}$, its selection probability $p_k$ is computed by normalizing over all merging scores (mentioned in Sec. 3.2) as:

$$
p_k = \frac{\exp\left(\langle \mathbf{q}, \mathbf{r}_k^{l+1} \rangle\right)}{\sum_{k=1}^{K} \exp\left(\langle \mathbf{q}, \mathbf{r}_k^{l+1} \rangle\right)}. \tag{9}
$$

As for $\pi_{\theta_\mathcal{C}}\left(\mathcal{C}_l \mid \mathbf{s}, \mathcal{M}_{<l+1}, \mathcal{C}_{<l}\right)$ in the check process, it follows a Bernoulli distribution with expectation $p_c^l = \sigma(\mathbf{W}_c \mathbf{r}_k^{l+1} + b_c)$, where $\theta_\mathcal{C} = \{\mathbf{W}_c, b_c\}$ are learned parameters. $p_c^l$ is indeed the trigger probability $p_c$ mentioned in Sec. 3.2.

Table 4: The dataset splits for all tasks.

| Dataset | SCAN | | | | | SCAN-ext | MiniSCAN |
|---|---|---|---|---|---|---|---|
| | Simple | Add Jump | Around Right | Length | MCD (1/2/3) | Extend | Limit |
| Train Size | 16728 | 14670 | 15225 | 16990 | 8365 | 20506 | 14 |
| Test Size | 4182 | 7706 | 4476 | 3920 | 1045 | 4000 | 8 |

**Low-level policy** When the high-level action $\mathcal{G}$ is determined, the low-level agent is triggered to output $\mathbf{a}$ according to the low-level policy $\pi_\varphi(\mathbf{a}\,|\,\mathcal{G},\mathbf{s})$. The policy $\pi_\varphi(\mathbf{a}\,|\,\mathcal{G},\mathbf{s})$ is implemented by an LSTM-based sequence to sequence network with an attention mechanism, i.e.,

$$\pi_\varphi(\mathbf{a}=(\mathrm{a}_1\cdots\mathrm{a}_M)\,|\,\mathcal{G},\mathbf{s})=\prod_{m=1}^{M}\pi_\varphi\left(\mathrm{a}_m\,|\,\mathcal{G},\mathbf{s},\mathrm{a}_{<m}\right), \qquad (10)$$

where $M$ is the number of decoding steps and $\mathrm{a}_m$ represents an action word (e.g. JUMP), or a destination variable (e.g. \$Y) which will be replaced by its corresponding constant DstExp stored in Memory. At each decoding step, $\mathrm{a}_m$ is sampled from a categorical distribution, whose sample space consists of all action words and destination variables with non-empty value slots.

## C   Chain Rule Derivation

Looking back to Eq. 2, the parameters $\theta$ and $\varphi$ can be optimized by ascending the following gradient:

$$\nabla_{\theta,\varphi}\mathcal{J}(\theta,\varphi)=\mathbb{E}_{\tau\sim\pi_{\theta,\varphi}}\,R(\tau)\nabla_{\theta,\varphi}\log\pi_{\theta,\varphi}\left(\tau\right), \qquad (11)$$

where the policy $\pi_{\theta,\varphi}$ can be further decomposed into a sequence of actions and state transitions:

$$\begin{aligned}\pi_{\theta,\varphi}(\tau)&=p(\mathbf{s}^1\mathcal{G}^1\mathbf{a}^1\cdots\mathbf{s}^T\mathcal{G}^T\mathbf{a}^T)\\&=p(\mathbf{s}^1)\prod_{t=1}^{T}\pi_{\theta,\varphi}(\mathbf{a}^t,\mathcal{G}^t\,|\,\mathbf{s}^t)\,p(\mathbf{s}^{t+1}\,|\,\mathbf{s}^t,\mathcal{G}^t,\mathbf{a}^t).\end{aligned} \qquad (12)$$

Consider that the low-level action $\mathbf{a}^t$ is conditioned on the high-level action $\mathcal{G}^t$, which means that $\pi_{\theta,\varphi}(\mathbf{a}^t,\mathcal{G}^t\,|\,\mathbf{s}^t)=\pi_\theta(\mathcal{G}^t\,|\,\mathbf{s}^t)\pi_\varphi(\mathbf{a}^t\,|\,\mathcal{G}^t,\mathbf{s}^t)$, and thus $\pi_{\theta,\varphi}(\tau)$ can be expanded as:

$$\pi_{\theta,\varphi}(\tau)=p(\mathbf{s}^1)\prod_{t=1}^{T}\pi_\theta(\mathcal{G}^t|\mathbf{s}^t)\pi_\varphi(\mathbf{a}^t|\mathcal{G}^t,\mathbf{s}^t)p(\mathbf{s}^{t+1}|\mathbf{s}^t,\mathcal{G}^t,\mathbf{a}^t). \qquad (13)$$

Since the state at step $t+1$ is fully determined by the state and actions at step $t$, not dependent on the policy parameters $\theta$ and $\varphi$, the gradients of $p(\mathbf{s}^{t+1}\,|\,\mathbf{s}^t,\mathcal{G}^t,\mathbf{a}^t)$ and $p(\mathbf{s}^1)$ with respect to $\theta$ and $\varphi$ are 0. Therefore, $\nabla_{\theta,\varphi}\mathcal{J}(\theta,\varphi)$ can be expanded as:

$$\begin{aligned}\nabla_{\theta,\varphi}\mathcal{J}(\theta,\varphi)&=\mathbb{E}_{\tau\sim\pi_{\theta,\varphi}}\,R(\tau)\nabla_{\theta,\varphi}\log\pi_{\theta,\varphi}(\tau),\\&=\mathbb{E}_{\tau\sim\pi_{\theta,\varphi}}\,R(\tau)\nabla_{\theta,\varphi}\sum_{t=1}^{T}\left[\log\pi_\theta\left(\mathcal{G}^t|\mathbf{s}^t\right)+\log\pi_\varphi\left(\mathbf{a}^t|\mathcal{G}^t,\mathbf{s}^t\right)\right],\\&=\mathbb{E}_{\tau\sim\pi_{\theta,\varphi}}\,R(\tau)\sum_{t=1}^{T}\left[\nabla_{\theta,\varphi}\log\pi_\theta\left(\mathcal{G}^t|\mathbf{s}^t\right)+\nabla_{\theta,\varphi}\log\pi_\varphi\left(\mathbf{a}^t|\mathcal{G}^t,\mathbf{s}^t\right)\right].\end{aligned} \qquad (14)$$

## D   Data Splits

As for data splits, we split each dataset into the train set and the test set for all tasks according to previous works. More details about train and test sizes can be seen in Tab. 4. More specifically, except for the task *Limit*, we further randomly take $20\%$ training data as the development set to tune the hyperparameters, with the rest being the train set.