[Reviews · NeurIPS 2020]

Review 1

Summary and Contributions: The paper proposes a system for achieving compositional generalization in sequence-to-sequence translation. The main contributions are as follows: 1) A new (to the best of my knowledge) approach to learning analytical expressions using hierarchical reinforcement learning, called LAnE. 2) Extensive evaluation on SCAN (Lake and Baroni 2018). The experiments include the standard systematicity and productivity tasks in SCAN as well as few-shot learning on MiniScan (Lake et al., 2019) and experiments on the distribution-based train-test splits (following the work on Maximum Compound Divergence by Keysers et al 2020). All of the reported results of the proposed system are 100%. 3) Ablation studies on SCAN which highlight the importance of different parts of the proposed system, including the results which show that without curriculum learning the performance drops to the test accuracy of 0% across all tasks.

Strengths: - Work at the intersection of reinforcement learning and natural language is of high interest to the community. Similarly, the topic of compositional generalization in language models is of high interest. - The authors introduce new extensions to SCAN: generalization to longer test sequences (SCAN-ext), distribution-based systematic generalization, curriculum learning.

Weaknesses: - The authors should clarify what they mean by the "extra resources" when positioning their work in comparison to the existing methods. The Memory part of LAnE and the use of curriculum learning (without which the accuracy drops from 100% to 0%) can be argued to be an additional resource, a similar one to permutation-based augmentation in Meta Seq2Seq (Lake 2018). More information on what distinguishes curriculum learning (which lead to the high performance of LAnE) from permutation-based augmentation would strengthen the paper. - The ablation studies in Table 3 are quite puzzling when it comes to the importance of curriculum learning. Why is test accuracy decreasing from 100% (averaged across 5 independent runs) to 0% without curriculum learning, across all tasks? The authors designed the curriculum based on the sequence length: the longer sequences are considered more difficult. What happens in the case of other curricula designs, for instance, if the difficulty of a sample is approximated using MCD or other metrics related to compositional generalization? Experimental results across all tasks rely on curriculum learning (Table 3), while there is no discussion on why this is the case. The unexplained significance of curriculum learning in achieving high results with the proposed system is my main criticism when it comes to this submission.

Correctness: Based on my understanding of Sections 3-5, empirical methodology is correct. Some of the claims are not justified: for instance, line 293 "Meanwhile, LAnE does not rely heavily on a particular combination of learning rates, suggesting its robustness" does not correspond to Figure 5, which shows that train and test accuracies are sensitive to the learning rate combination.

Clarity: Methodology section is hard to follow. Concepts should be explained as soon as they are introduced: for example, line 121 mentions merging score which is not defined. Then in line 127 it is defined using another new concept of a "learnable query vector q" which is then left unexplained. Appendix does not explain what q is either. I suggest explaining the Figure 1 in the caption instead of captioning the figure with "see text". Can you explain if the Composer finds recognizable SrcExp in an order-invariant way with respect to the input words? The example "$x after $y” is said to trigger two different behaviours in line 116 (recognize “$x after $y” and trigger Solver) and line 150 (recognize "$y twice" before “$x after $y” in the “$x after $y twice” sequence).

Relation to Prior Work: There is a section on related work, however, more clarity regarding "additional resources" and a comparison of the employed curriculum learning with perturbation-based augmentation in Meta Seq2Seq would strengthen the paper.

Reproducibility: No

Additional Feedback:


Review 2

Summary and Contributions: This paper introduces a novel neuro-symbolic approach to compositional generalization. The proposed architecture involves two neural models working together that parse input instructions (composer) and interpret the pieces with learned rules (solver). The model has a built-in notion of rules and variables, although the rules themselves are learned. The model is trained with reinforcement learning. This is a very strong paper. I thoroughly enjoyed reading about your work. I would like to see this paper published at NeurIPS.

Strengths: Here are some of the strong points: - The model is a genuinely novel approach to compositional generalization and SCAN - The model combines some of the best of previous approaches---abstract rules and neural networks---without the need for strong supervision regarding the underlying rules or how they are used. - The paper also introduces a "SCAN-ext" dataset that further probes productivity through recursive conjunctions, which will be useful in evaluating future models. - The results are striking - Surprisingly, the model even solves the small MiniSCAN dataset perfectly - Ablation studies and thorough comparisons are provided - The paper is well-written

Weaknesses: I have only a two suggestions. Both are readily addressable in a revision. -The paper is well-written, but I still had to read the model section a few times (and study the figures very carefully) to understand how it works. My suggestion is to provide clearer description of the complete model at a high-level (like Section 3.2 "Interaction") before diving into the sections on the specific pieces. Currently, the first paragraph in Section 3.2 aims to do this, but a better description is needed at that point in the ppaer. - I would also like to know more about the rules the model learns. Figure 6 provides some hint of this, but the paper does not mention whether the model learns anything like the rules in Figure 1. Also, how do the learned solver rules relate to the actual interpretation rules that were used to define meaning in SCAN? Or the SCAN phrase structure grammar that defines the space of commands?

Correctness: Yes

Clarity: Yes

Relation to Prior Work: Yes

Reproducibility: Yes

Additional Feedback: To recap, this is a great paper and a worthy contribution to NeurIPS. I will advocate for its acceptance. Minor comments: - It would be nice if you explicitly mention what your model assumes as prior knowledge, and what is learned -pg. 2 : "we believe our work could shed light on the community". This is an odd sentence, and I don't it says what you mean. - pg. 3 : "from bottom to up" -> "From bottom to top" --- After rebuttal --- Thank you for addressing my questions in the rebuttal. The authors promised to make substantial revisions to improve clarity, and I trust they will make these improvements in a revision. I thought this was a strong paper in the first round, and I will keep my score.


Review 3

Summary and Contributions: The paper proposes a neuro-symbolic algorithm for compositional generalization in seq-2-seq problems. The algorithm solves seq to seq problems by using neural modules to find and apply analytical expressions. The method is trained via hierarchical reinforcement learning, and achieves 100% accuracy on the SCAN and MiniSCAN domains.

Strengths: The system architecture is novel, and the empirical results are strong. In particular, it was surprising to see strong performance on the MiniSCAN test problem (which only contains 14 training examples) without meta-learning or program synthesis. The ablations provide good analysis of the importance of the individual components of the system. Significance: The problem of compositional generalization has seen much attention recently, and this work provides a neuro-symbolic framework to tackle this problem.

Weaknesses: I think the main weakness of this paper is its clarity. See below for discussion. This work could also benefit from analysis in more domains, to show that it is applicable beyond the SCAN/MiniSCAN framework. See below for discussion.

Correctness: The claims and methodology seems correct.

Clarity: I think the paper suffers from a major lack of clarity. I had to read the paper (particularly section 3) a few times to understand how the model works and what the main results were, which is unfortunate, because I did find the approach and results interesting once I understood. Some notes: - I found Figure 2 (the key model figure) hard to interpret (see below for suggestions). - The operation of the composer and the notion of recognizable expressions was difficult to understand, I think these terms could be defined more clearly, and perhaps shown explicitly in a figure. - Thoroughly proofreading the paper for style and grammar would greatly increase readability.

Relation to Prior Work: I think related work is generally discussed well. In Table 1 and Sec 5, I don’t think it is appropriate to say “no extra resources used” for some methods but not others; the choices made in the design of the individual components, the reward function, and the training curriculum all provide important inductive biases beyond a “naive” approach.

Reproducibility: Yes

Additional Feedback: I think that the results on the few-shot MiniSCAN problem are very interesting and worth emphasizing more than they are emphasized currently. It’s surprising and impressive that this approach can learn to generalize from such a limited dataset. I would definitely recommend finding more domains/problems which can show how well this approach works in the few-shot setting, because advances there would have very high value. Evaluation on more domains (particularly naturalistic ones) would also be helpful in showing that this approach can help solve compositional generalization problems in real-world settings. For instance, Nye et al (2020) show that the program synthesis approach can be applied to learn how to interpret number words in novel language from a limited number of examples; it would be really interesting to see if this approach could be applied to that problem or similar real-world problems. (Again, particularly in the few-shot setting) Regarding clarity of Figure 2: For neuro-symbolic methods, I think it is important to be very explicit in the diagram, caption, mathematical notation, and text about which components are neural networks, which components are symbolic machines, and which components are vectors/symbols. I don’t think Figure 2 or the Figure 2 caption does a good job of this. The caption could also explicitly take the reader through the execution of the machine. If you modified Figure 2 so that looking at the figure and reading the caption gave a clear picture of how the overall system works, it would be a huge help for understanding section 3. Overall, I think this is promising work which suffers from a lack of clarity in the presentation. I think rewriting the paper would make it strong. M. I. Nye, A. Solar-Lezama, J. B. Tenenbaum, and B. M. Lake. Learning compositional rules via neural program synthesis. 2020. ------ after author response: ------ Thank you for the response, which discussed the concerns regarding clarity with concrete proposals, and also addressed my other comments about extra resources and additional domains. Given this, I have raised my score.


Review 4

Summary and Contributions: LANE is a novel framework developed based on the SCAN task, which is a semantic parsing task specifically developed for measuring the compositional generalization abilities of learning systems based on various holdout splits. LANE is based on a Composer, a Solver, and Memory. The Composer uses a TreeLSTM to find subexpression in the input that can be mapped to output expressions. These subexpressions are then translated into output expression by the Solver, which is a seq2seq LSTM with attention. Finally, Memory is used to remember the mappings in variables, which allows them to be replaced at later time points. The whole architecture is learned end-to-end using Hierarchical Reinforcement Learning. The reward function is designed carefully, and ablation studies show that using curriculum learning turns out to be one of the most important features of the architecture. The empirical results show perfect scores on the traditional SCAN splits and the SCAN MCD split. It is the first time someone is able to achieve this.

Strengths: In my opinion this is one of the most innovative approaches that has been proposed based on SCAN up to now. The way the various techniques are put together is highly non-trivial. The empirical results are very strong: the SCAN dataset is popular, but up to now nobody has been able to get to perfect performance on all traditional splits (including the length split) without additional side-input in the form of primitive substitutions or grammar rules. Moreover, the SCAN MCD split is even more challenging, and they are able to solve that perfectly as well.

Weaknesses: Since the framework is quite complex and designed specifically for SCAN, it is not clear how it generalizes to other datasets such as the Compositional Freebase Questions. Recent research [1] shows that strong performance on SCAN splits does not necessarily mean it will do well on other datasets as well. I'd be interested in understanding better how the authors plan to apply their framework to CFQ. Furthermore, the authors provide no information on how long it took to train the models and how much / which resources were need. This should be added to the paper. [1] https://arxiv.org/pdf/2007.08970.pdf

Correctness: I am not an expert in Hierarchical Reinforcement Learning, so I can't say whether all derivations used there are correct. The remaining claims, method and empirical methodology all seems correct to me.

Clarity: Yes.

Relation to Prior Work: Yes, the authors compare their approach to the most relevant recent contributions, and their idea of separating standard baselines from those that use additional resources is a good idea.

Reproducibility: Yes

Additional Feedback: === Overal === * Please add more details on your experimental setup. How long did it take to train? What devices did you use? How many replications did you run? * Curriculum learning is absolutely crucial to your approach, but you don't provide much insight why. I suppose this is because you are using reinforcement learning, so for long sentences it is more difficult to assign credit. Or do you have some other explanation? * In Section 5.2 you claim "LANE is capable of learning human prior knowledge to some extent". Could you clarify this? What kind of human prior knowledge are you referring to? === Typos === Section 1: * You are claiming that reference [10] evaluated a Transformer on SCAN. This is incorrect; they only evaluated a CNN. Section 2: * "the compositional generalization" --> remove "the" * "MCD[17]" --> add space Section 3: * "as a source" --> a source " "as a destination" -> a destination * "From bottom to up" --> from bottom to top === After rebuttal === I'm satisfied with the authors' clarification on curriculum learning and details on the experimental setup.

[Author Response · NeurIPS 2020]

We really appreciate all reviewers for their careful reviews and constructive comments. We are pleased that all reviewers
find our work innovative and interesting. In the following, we highlight the primary issues and questions proposed by
the reviewers. Remaining minor comments and suggestions will also be incorporated in the revision to polish our paper.

## Primary Issues

**The importance of curriculum learning (R1, R4)**. Thank R1 and R4 for pointing it out. As supposed by R4, the main
reason why curriculum learning seems to be so important is that our model uses reinforcement learning. It is not easy to
make the model practically effective since its action space is exponentially large, which is due to the indefinite length of
output sequences in Solver, and the huge number of possible trees in Composer. Such a huge space means that rewards
are very sparse, especially for harder examples. Without curriculum learning, our randomly initialized model receives
zero rewards on most examples and shows no sign of convergence even after training for several days. Therefore, test
(even train) accuracies in Table 3 are zero across all tasks. In comparison, by arranging examples from easy to hard,
curriculum learning alleviates the sparse reward issue. On the one hand, easy examples are more likely to provide
non-zero rewards to help our model converge; on the other hand, models trained on easy examples have a greater
possibility to receive non-zero rewards on hard examples. Regarding R1's consideration about different curriculum
settings, we have not tried others since we find the simplest length-based curriculum effective enough. We think other
settings such as MCD-based curriculum may also work. We will include the above explanation in the revision.

**The clarification of "extra resources" (R1, R3)**. We clarify "extra resources" here as "data or data specific rules
other than original training data". GECA and Meta Seq2Seq lie in "extra resources" since they both utilize extra data.
Specifically, GECA recombines real examples to construct extra data, while Meta Seq2Seq employs random assignment
of the primitive instructions (e.g. "jump") to their meaning (e.g. JUMP) to synthesize extra data. Regarding data-specific
rules, Equivariant Seq2Seq requires predefined local groups to make the model aware of equivariance between verbs
or directions (line 321, e.g. "jump" and "run" are verbs). Similarly, Program Synthesis needs a predefined meta
grammar, which heavily relies on the grammar of a dataset, and hence we think it also falls into the group of using
"extra resources". In comparison, other methods including ours (Table 1) do not utilize any synthetic data or data
specific rules. We will define "extra resources" explicitly in the revision.

**The representation of methodology section (R1, R2, R3)**. Thanks for all the kind suggestions, and we will carefully
proofread the section to achieve better readability in the revision. First, we will give a clear picture of how the overall
system works at the beginning of Section 3, and explicitly mark the neural parts and symbolic parts. Second, we will
utilize two figures instead of Figure 2 to illustrate the overall system and the detail of Composer respectively. In order
to make the figures easily understood, we will polish all the captions with enough information. Last, we will invite
native speakers to help us thoroughly proofread our paper for grammar and style.

## Other Questions

**Explain the behavior on "$x after $y twice" (R1)**. In fact, our model does not depend on a specific order when
finding the recognizable SrcExp. The different behaviors in R1's cases are normal since the finding process is context
sensitive. Taking "$x after $y twice" as an example, our model will assign the highest merging score on "$y twice",
which means that it will first merge "$y twice'' in the context of "$x after $y twice".

**What kind of grammar does the model learn (R2)**. We find that the grammar learned by our model is very similar to
the phrase-structure grammar defined in SACN. Regarding the difference, because of our designed simplicity-based
reward, our model tends to learn rules such as "D -> X Y", "Y -> left" and "Y -> right", instead of "D -> U left" and
"D -> U right" in the SCAN grammar. These two kinds of rules are both valid on SCAN. And we will provide more
examples for better understanding the learnings of our model.

**More analysis on MiniSCAN and real-world settings (R3)**. We also think compositional generalization on few-shot
tasks and real-world applications are interesting. We will follow R3's advice to complement more analysis on our
MiniSCAN experiments. As for real-world applications, one of our future work is to experiment on the dataset CFQ.

**Extend this method to CFQ (R4)**. Although still under progress, we believe that with some modifications on Solver
(i.e., switching from "NL to actions" to "NL to SPARQL"), our method can readily generalize to the CFQ dataset.

**Details about training devices and time (R4)**. Our model was trained on a single Tesla-P100 (16GB) and the training
time for a single run is about $20 \sim 25$ hours. We will add it to our paper in the revision. For each result of our method,
we run 5 replications, as indicated in the caption of Table 1.

**What kind of human prior knowledge is learned (R4)**. The prior knowledge here is referred to the phrase-structure
grammar defined in SCAN. This claim is based on the observation that our model can achieve perfect performance
without injecting such knowledge manually, indicating that our model learns this knowledge to some extent.

[Meta-Review · NeurIPS 2020]

This paper proposes a neuro-symbolic method (LAnE) for sequence-to-sequence tasks that exhibits systematicity and can solve both SCAN and MiniSCAN with perfect accuracy. LAnE works by identifying subexpressions in the input, translating them to the output domain (leveraging previously translated subexpressions stored in memory), and finally updating the outputs in memory. The reviewers found the paper to be of high quality, praising its novelty, the strength of the results, and the interest it is likely to generate amongst other researchers. In particular, R2 noted the “genuinely novel approach” and that the “results are striking”; similarly R4 stated that the paper presents “one of the most innovative approaches that has been proposed”. The biggest concern amongst the reviewers was clarity, with R2 and R3 mentioning they had to read the methods section several times and R1 also mentioning it was hard to follow. However, the reviewers were satisfied by the authors’ response in which they promise several concrete proposals to improve that section. I agree that the paper’s results are very strong and believe it will be of interest to the NeurIPS community. I therefore recommend acceptance.